# Synchronization Induced by Layer Mismatch in Multiplex Networks

**DOI:** 10.3390/e25071083

**Published:** 2023-07-19

**Authors:** Md Sayeed Anwar, Sarbendu Rakshit, Jürgen Kurths, Dibakar Ghosh

**Affiliations:** 1Physics and Applied Mathematics Unit, Indian Statistical Institute, 203 B. T. Road, Kolkata 700108, India; sayeed_r@isical.ac.in; 2Department of Mechanical Engineering, University of California, Riverside, CA 92521, USA; sarbendu.rakshit@ucr.edu; 3Potsdam Institute for Climate Impact Research, Telegraphenberg A 31, 14473 Potsdam, Germany; juergen.kurths@pik-potsdam.de; 4Department of Physics, Humboldt University Berlin, 12489 Berlin, Germany

**Keywords:** synchronization, multiplex networks, parameter mismatch, master stability function

## Abstract

Heterogeneity among interacting units plays an important role in numerous biological and man-made complex systems. While the impacts of heterogeneity on synchronization, in terms of structural mismatch of the layers in multiplex networks, has been studied thoroughly, its influence on intralayer synchronization, in terms of parameter mismatch among the layers, has not been adequately investigated. Here, we study the intralayer synchrony in multiplex networks, where the layers are different from one other, due to parameter mismatch in their local dynamics. In such a multiplex network, the intralayer coupling strength for the emergence of intralayer synchronization decreases upon the introduction of impurity among the layers, which is caused by a parameter mismatch in their local dynamics. Furthermore, the area of occurrence of intralayer synchronization also widens with increasing mismatch. We analytically derive a condition under which the intralayer synchronous solution exists, and we even sustain its stability. We also prove that, in spite of the mismatch among the layers, all the layers of the multiplex network synchronize simultaneously. Our results indicate that a multiplex network with mismatched layers can induce synchrony more easily than a multiplex network with identical layers.

## 1. Introduction

The study of complex network theory provides a rich foundation for comprehending the inherent characteristics and various emergent dynamics within interacting units [1,2,3]. Over the years, network science has garnered significant interest, due to its practical importance in modeling a variety of biological, physical, engineering, and social systems that are prevalent in both society and nature [4,5,6]. In order to achieve optimal performance, the constituent units of a network may engage in interactions with other networks through a variety of coupling mechanisms and interacting patterns. These interactions give rise to a range of emergent phenomena. Recent studies have uncovered a complex network structure, called multilayer networks [7,8,9], that encompasses the structural characteristics of diverse real-life systems within a unified network framework, providing an accurate portrayal of numerous interacting complex systems found in nature [10,11,12,13]. A particular instance of a multilayer network is a multiplex network [14,15,16], wherein each layer comprises an equal number of constituent units (nodes) that engage in interactions via intralayer links. Additionally, a node from a given layer is interconnected with its twin node in all the other layers via interlayer connections.

As the elementary units of a network are usually dynamical systems, subjected to the impact of the states of other connected units, it is crucial to investigate the interplay between the structure and underlying dynamical process of a network [9]. The impact of these distinct characteristics within networked systems results in the identification of numerous compelling emergent phenomena [17,18,19]. Synchronization [20,21,22,23,24,25,26,27], wherein the system individuals evolve in unison, is one of the fascinating phenomena observed in dynamical networks, which have attracted a lot of attention in the area of network research.

Diverse forms of synchronization patterns have been detected in multiplex networks with static or temporal interactions, including—but not limited to—cluster synchronization [28,29], explosive synchronization [30], chimera states [31], intralayer synchronization [32,33], interlayer synchronization [34,35,36], and relay synchronization [37,38,39]. Recently, the study of synchronization in multilayer networks has also been extended to include higher-order interactions [36,40]. In this context, most of these studies on synchronization have been performed by assuming networks of coupled homogeneous oscillators. Only a few have considered networks of coupled oscillators with heterogeneity, a common occurrence in natural systems [41,42,43,44,45,46,47,48,49]. A study conducted by Plotnikov et al. [45] revealed that the expansion of heterogeneity within a network of heterogeneous FitzHugh–Nagumo neurons results in improvement of synchrony. Gambuzza et al. [32] analyzed intralayer synchronization in a multiplex network comprising two distinct layers of oscillators, where one layer consisted of detached oscillators, while the other one comprised interconnected oscillators with a mismatch in intrinsic frequencies. Leyva et al. [34] studied the interlayer synchronization in multiplex networks with nonidentical connectivity structures of the layers. Rakshit et al. [41] conducted a study on the relay interlayer synchronization in multiplex networks that included an impurity, specifically a parameter mismatch in the local dynamics of the nodes within the relay layers. Despite this, there are many facets of the relationship between multilayer structure and heterogeneity in oscillators that have not been thoroughly investigated. In particular, the research of intralayer synchronization in multiplex networks with heterogeneous layers is in its infancy.

In this article, we try to fill this gap and investigate the intralayer synchrony in multiplex networks in which the layers are heterogeneous from one other, by means of parameter mismatch in their local dynamics, although the local dynamics of each node within a particular layer are identical. Under such broad circumstances, we derive a condition for the existence of an intralayer synchronous solution. Using the master stability function approach [50,51], we acquire the necessary condition for this solution to be stable, and we show that for some particular instances, the problem of stability can be transformed into a simplified form. We further demonstrate that even though the layers of the multiplex network are not identical, they exhibit simultaneous synchronization. We investigate the effect of parameter heterogeneity on the emergence of intralayer synchronization. Our study demonstrates that the layers can attain synchrony at relatively reduced levels of coupling strengths, as the mismatch parameter between the layers increases, and that the area of intralayer synchronization expands. These findings indicate that the introduction of a mismatch between layers, by altering parameter values in one layer, can enable synchronization in the other layer at coupling strengths where synchronization would not occur if the layers were identical.

## 2. Mathematical Model

We start by considering a *Q*-layered multiplex network, each of these layers being composed of *N* nodes of *d*-dimensional dynamical systems. The layers are made of a fixed set of nodes, and interlayer connections exist between each node of a layer and all its replicas in the adjacent layers, where the layers are organized in a chain. Then, the mathematical form of the entire dynamical multiplex network can be described as
(1)$$\begin{array}{c} {\dot{V}}_{m,i}=F\left(V_{m,i},\phi_{m}\right)+\sigma_{m}\sum_{k=1}^{N}C_{ik}^{\left(m\right)}G\left(V_{m,i},V_{m,k}\right)+\sigma_{inter}\Gamma \sum_{p=m-1>0}^{p=m+1\leq Q}\left[V_{p,i}-V_{m,i}\right], \end{array}$$
where i=1,2,…,N; m=1,2,…,Q. Here, the states of the *i*th node in layer-*m* are represented by the *d*-dimensional state vector $V_{m,i}$. The dynamics of each individual oscillator are governed by the continuously differentiable evolution function $F:R^{d}\times R\to R^{d}$. We keep the vector field *F* as identical for all subsystems in a particular layer, whereas it is different for different layers, in terms of mismatch in the system parameter $\phi_{m}$. The interlayer inner-coupling matrix Γ expresses the coupling connections between the layers. *G* is the intralayer coupling function, which represents the coupling connections within the layers. The interaction between the nodes within layer-*m* is controlled by $\sigma_{m}$, which is the intralayer coupling strength of layer-*m*. For the sake of simplicity, we consider the intralayer coupling strength to be equal in each layer, given by $\sigma_{m}=\sigma_{intra}$, for all *m*. On the other hand, how the information will be conveyed between the layers is determined by the parameter $\sigma_{inter}$, called the interlayer coupling strength. Figure 1 represents a schematic diagram of a two-layer multiplex network, where the local dynamics of the nodes in a particular layer are identical to one other, while the local dynamics of the nodes in two different layers are different from one another, in terms of mismatch in the parameter values of the local dynamics.

The intralayer network configuration of layer-*m* is encoded by the *N*-order adjacency matrix $C^{\left(m\right)}$, which describes the interconnections between individual oscillators in layer-*m*. Here, $C_{ij}^{\left(m\right)}=1$ if the *i*-th and *j*-th node of layer-*m* are connected, and is equal to 0, otherwise. We consider the connection between the *i*-th and *j*-th nodes of layer-*m* to be undirected, so that the resulting adjacency matrix $C^{\left(m\right)}$ is real symmetric. From the adjacency $C^{\left(m\right)}$, one can obtain the corresponding zero-row-sum real-symmetric Laplacian matrix $L^{\left(m\right)}$, defined as $L_{ij}^{\left(m\right)}=-C_{ij}^{\left(m\right)}$ if i≠j, and $L_{ii}^{\left(m\right)}=\sum_{j=1}^{N}C_{ij}^{\left(m\right)}$: that is, the off-diagonal elements are the negative of the corresponding elements in $C^{\left(m\right)}$, and the diagonal element $L_{ii}^{\left(m\right)}$ is the sum of the non-diagonal elements in the *i*-th row of $C^{\left(m\right)}$, which is basically the degree $k_{i}^{\left(m\right)}=\sum_{j=1}^{N}C_{ij}^{\left(m\right)}$ of the *i*-th node in Layer-*m*.

## 3. Results

Throughout this section, we will investigate the occurrence of a particular synchronization phenomenon, called intralayer synchronization, in our multiplex framework (Equation 1). To do so, we first analytically derive the necessary conditions for the existence and stability of the intralayer synchronous solution.

### 3.1. Analytical Results

The multiplex network (Equation 1) achieves an intralayer synchronization state when all the nodes in a particular layer converge to the same time evolution. Then, for layer-*m*, there exists a solution $V_{m,s}\left(t\right)\in R^{d}$, such that $∥V_{m,i}\left(t\right)-V_{m,s}\left(t\right)∥\to 0$ as t→∞ for all i=1,2,…,N. Thus, all the nodes in a particular layer-*m* converge on one other, i.e., asymptotically $V_{m,i}\left(t\right)\to V_{m,j}\left(t\right)$ as t→∞, although they may have separate trajectories, layer-wise. Consequently, the intralayer synchronization manifold $S^{intra}$ can be defined as
(2)$$\begin{array}{c} S^{intra}=\{\left(V_{1,s}\left(t\right),V_{2,s}\left(t\right),\ldots ,V_{Q,s}\left(t\right)\right)\subset R^{Qd}:\\ V_{m,i}\left(t\right)=V_{m,s}\left(t\right),\;i=1,2,\ldots ,N;\;m=1,2,\ldots ,Q;\;t\in R^{+}\}. \end{array}$$

#### 3.1.1. Invariance Condition

The proposed multiplex network is not always capable of reaching an intralayer synchronous state, because the presence of arbitrarily intralayer connection topologies and intralayer coupling functions between the unitary dynamical units does not ensure that the unitary components of a particular layer will progress in unison. Therefore, we take into account a certain form of the intralayer coupling functions that disappears at the referred synchronous state, i.e., $G(V_{m,s},V_{m,s})=0$, for all *m*. This particular type of coupling function is called “synchronization noninvasive couplings” [40,52]. If the intralayer coupling functions are synchronization noninvasive, then the existence and invariance of an intralayer synchronous solution can be guaranteed in the multiplex network (Equation 1), because each unitary component of a particular layer follows the same evolution dynamics given by
(3)$$\begin{array}{c} {\dot{V}}_{m,s}=F\left(V_{m,s},\phi_{m}\right)+\sigma_{inter}\Gamma \sum_{p=m-1>0}^{p=m+1\leq Q}\left[V_{p,s}-V_{m,s}\right],\;\;m=1,2,\ldots ,Q. \end{array}$$
One of the benefits of selecting this particular type of coupling lies in its ability to accommodate any arbitrary connection topology between the unitary components within the layers. Furthermore, it is worth mentioning that our approach, of incorporating synchronization noninvasive coupling functions, covers a wide range of coupling schemes, including various types of coupling functions, such as generalized diffusive coupling functions, electrical synaptic couplings in neuronal networks, and the diffusive sine couplings used in coupled Kuramoto oscillators.

Additionally, for the emergence of the intralayer synchronous solution, it is possible to choose an arbitrary type of coupling function other than the synchronization noninvasive couplings. However, in this case, to ensure the constancy of the intralayer synchronous solution, it is necessary to relinquish the unrestricted selection of connection topologies, and to exclusively choose regular connectivity configurations [33]. Nevertheless, in the present study, we will only consider the case of synchronization noninvasive intralayer coupling functions, which guarantee the invariance of an intralayer synchronous solution with an arbitrary connectivity structure within the layers.

#### 3.1.2. Stability Analysis

Now, the question is whether the intralayer synchronized state sustains its stability or, conversely, becomes unstable, in response to small perturbations. In order to conduct an inquiry into this matter, we apply small variations to the synchronous solution, defined as $\delta V_{m,i}=V_{m,i}-V_{m,s}$, (i=1,2,…,N) and (m=1,2,…,Q). The dynamical Equation (Equation 1) is then linearized, by substituting the expressions for perturbations $\delta V_{m,i}$, and utilizing the Taylor series expansion up to the linear order, which eventually returns the variational equation, in terms of perturbations, as
(4)$$\begin{array}{c} \delta {\dot{V}}_{m,i}=JF\left(V_{m,s},\phi_{m}\right)\delta V_{m,i}+\sigma_{intra}\sum_{k=1}^{N}C_{ik}^{\left(m\right)}\left[J_{1}G\delta V_{m,i}+J_{2}G\delta V_{m,k}\right]\\ \;\;\;\;\;\;\;\;\;\;\;\;\;\;\;\;\;\;\;\;\;\;\;\;\;\;\;\;\;\;\;\;\;\;\;+\sigma_{inter}\Gamma \sum_{p=m-1>0}^{p=m+1\leq Q}\left[\delta V_{p,i}-\delta V_{m,i}\right], \end{array}$$
where $JF(V_{m,s},\phi_{m})$ is the Jacobian matrix of the function *F*, evaluated at the synchronous solution $V_{m,s}$, and where $J_{1}G$ and $J_{2}G$ are the partial derivatives of *G*, with respect to the first and second variables, respectively, and are evaluated at $\left(V_{m,s},V_{m,s}\right)$. The linearized Equation (Equation 4) can be further simplified, using the assumption of the synchronization noninvasive intralayer coupling function. As *G* vanishes at the synchronous manifold, i.e., its value is equal to 0, it follows that the total derivative of *G* also vanishes at the synchronous manifold. Hence, we have
(5)$$\begin{array}{c} J_{1}G+J_{2}G=0. \end{array}$$
Using the relation (Equation 5) and the definition for Laplacian matrices, one can rewrite the linearized Equation (Equation 4) as follows:(6)$$\begin{array}{c} \delta {\dot{V}}_{m,i}=JF\left(V_{m,s},\phi_{m}\right)\delta V_{m,i}-\sigma_{intra}\sum_{k=1}^{N}L_{ik}^{\left(m\right)}J_{2}G\delta V_{m,k}+\sigma_{inter}\Gamma \sum_{p=m-1>0}^{p=m+1\leq Q}\left[\delta V_{p,i}-\delta V_{m,i}\right]. \end{array}$$
We now rewrite the linearized Equation (Equation 6) in a block form, by introducing the stack vector $\delta V_{m}={\left[\delta V_{m,1}^{T},\delta V_{m,2}^{T},\ldots ,\delta V_{m,N}^{T}\right]}^{T}$, (m=1,2,…,Q), yielding
(7)$$\begin{array}{c} \delta {\dot{V}}_{m}=\left[I_{N}⊗JF\left(V_{m,s},\phi_{m}\right)-\sigma_{m}L^{\left(m\right)}⊗J_{2}G\right]\delta V_{m}+\sigma_{inter}\left(I_{N}⊗\Gamma \right)\sum_{p=m-1>0}^{p=m+1\leq Q}\left[\delta V_{p}-\delta V_{m}\right]. \end{array}$$

The above variational Equation (Equation 7) contains both the parallel and transverse components of the intralayer synchronization manifold. A stable synchronous solution can be achieved when all the transverse modes asymptotically die out. Hence, to decouple the transverse modes from the parallel one, we use the following conceptual steps associated with the Laplacian matrices of the layers: (i) as all Laplacian matrices $L^{\left(m\right)}$ are real symmetric, they are orthonormally diagonalizable by the associated set of eigenvectors $E^{\left(m\right)}=\left\{e_{1}^{\left(m\right)},e_{2}^{\left(m\right)},\ldots ,e_{N}^{\left(m\right)}\right\}$; (ii) due to the zero row sum property, all the Laplacian matrices share the common least eigenvalue $\lambda_{1}^{\left(m\right)}=0$, with the associated eigenvector $e_{1}^{\left(m\right)}=\frac{1}{\sqrt{N}}{\left(1,1,\ldots ,1\right)}^{T}$ that is aligned along the synchronization manifold; (iii) due to the non-commutativity property of the Laplacians, the eigenvector sets associated with distinct nonzero eigenvalues $\left(\lambda_{i}^{\left(m\right)}>0,i=2,3,\cdots ,N\right)$ are generally different from one other. However, it is noteworthy that any perturbation of the synchronization manifold can be expressed as a linear combination of one of these eigenvector sets. This implies that we can freely choose any of these layer-wise Laplacians as the reference for the basis of the transverse space, and all other eigenvector sets can be transformed into this basis through unitary matrix transformations.

Therefore, without loss of generality, we take the set of eigenvectors associated with the Laplacian matrix of layer-1 as the basis of reference, and we consequently introduce the new variables $\xi^{\left(m\right)}={\left(E^{\left(1\right)}⊗I_{d}\right)}^{-1}\delta V_{m}$. This eventually returns a variational equation in the following form:(8)$$\begin{array}{c} {\dot{\xi }}^{\left(m\right)}=\left[I_{N}⊗JF\left(V_{m,s},\phi_{m}\right)-\sigma_{intra}\Xi^{\left(m\right)}⊗DG\right]\xi^{\left(m\right)}+\sigma_{inter}\left(I_{N}⊗\Gamma \right)\sum_{p=m-1>0}^{p=m+1\leq Q}\left[\xi^{\left(p\right)}-\xi^{\left(m\right)}\right], \end{array}$$
where
(9)$$\begin{array}{c} \Xi^{\left(m\right)}=E^{{\left(1\right)}^{-1}}L^{\left(m\right)}E^{\left(1\right)}=\left[\begin{array}{cc} 0 & 0_{1\times (N-1)}\\ 0_{(N-1)\times 1} & {\tilde{L}}^{\left(m\right)} \end{array}\right] \end{array}$$
is the transformed Laplacian matrix, having a null first row and first column, and ${\tilde{L}}^{\left(m\right)}$ is an N×N real symmetric matrix with constant entries. Hence, using the above relation (Equation 9), one can eventually decouple the dynamics of parallel and transverse modes, as follows:
(10a)$$\begin{array}{c} {\dot{\xi }}_{\|}^{\left(m\right)}=JF\left(V_{m,s},\phi_{m}\right)\xi_{\|}^{\left(m\right)}+\sigma_{inter}\Gamma \sum_{p=m-1>0}^{p=m+1\leq Q}\left[\xi_{\|}^{\left(p\right)}-\xi_{\|}^{\left(m\right)}\right]; \end{array}$$
(10b)$$\begin{array}{c} {\dot{\xi }}_{⊥}^{\left(m\right)}=\left[I_{N-1}⊗JF\left(V_{m,s},\phi_{m}\right)-\sigma_{m}{\tilde{L}}^{\left(m\right)}⊗DG\right]\xi_{⊥}^{\left(m\right)}+\sigma_{inter}\left(I_{N-1}⊗\Gamma \right)\sum_{p=m-1>0}^{p=m+1\leq Q}\left[\xi_{⊥}^{\left(p\right)}-\xi_{⊥}^{\left(m\right)}\right], \end{array}$$
where $\xi_{\|}^{\left(m\right)}$ represents the state of perturbation modes parallel to the synchronization manifold, and where the states of perturbation modes across the synchronization manifold are illustrated by $\xi_{⊥}^{\left(m\right)}$. Now, as $E^{\left(1\right)}$ diagonalizes the Laplacian $L^{\left(1\right)}$, then ${\tilde{L}}^{\left(1\right)}=diag\left\{\lambda_{2}^{\left(1\right)},\lambda_{3}^{\left(1\right)},\cdots ,\lambda_{N}^{\left(1\right)}\right\}$; therefore, one can further simplify the dynamics of the transverse modes as
(11)$$\begin{array}{c} {\dot{\xi }}_{{⊥}_{i}}^{\left(1\right)}=JF\left(V_{1,s},\phi_{1}\right)\xi_{{⊥}_{i}}^{\left(1\right)}-\sigma_{intra}\lambda_{i}^{\left(1\right)}DG\xi_{{⊥}_{i}}^{\left(1\right)}+\sigma_{inter}\Gamma \left[\xi_{{⊥}_{i}}^{\left(2\right)}-\xi_{{⊥}_{i}}^{\left(1\right)}\right],\\ {\dot{\xi }}_{{⊥}_{i}}^{\left(m\right)}=JF\left(V_{m,s},\phi_{m}\right)\xi_{{⊥}_{i}}^{\left(m\right)}-\sigma_{intra}\sum_{k=1}^{N}{\tilde{L}}_{ik}^{\left(m\right)}DG\xi_{{⊥}_{k}}^{\left(m\right)}+\sigma_{inter}\Gamma \sum_{p=m-1>0}^{p=m+1\leq Q}\left[\xi_{{⊥}_{i}}^{\left(p\right)}-\xi_{{⊥}_{i}}^{\left(m\right)}\right], \end{array}$$
where i=2,3,…,N and m=2,3,…,Q. These transverse error dynamics are the required master stability equation, as solving the above Equation (Equation 11) for the calculation of the Lyapunov exponents gives the condition for the emergence of stable intralayer synchronization. The stability of the referred state requires, as a necessary condition, the maximum of these transverse Lyapunov exponents to be negative.

Note that the master stability equation is not fully decoupled: rather, it is a Q(N−1)d-dimensional coupled equation. In general, this master stability equation cannot be further reduced to a lower dimensional form; therefore, the calculation of the maximum Lyapunov exponent becomes too expensive. However, there are a few instances in which this intricacy can be overcome, where the master stability equation can be reduced to (N−1) numbers of Qd-dimensional equations. The relevant instances are illustrated below:If the connectivity structure of all the layers is identical, then all of them share the same Laplacian matrix. Without loss of generality, we choose the Laplacian matrix to be $L^{\left(1\right)}$. Then, the master stability equation becomes
(12)$$\begin{array}{c} {\dot{\xi }}_{{⊥}_{i}}^{\left(m\right)}=JF\left(V_{m,s},\phi_{m}\right)\xi_{{⊥}_{i}}^{\left(m\right)}-\sigma_{intra}\lambda_{i}^{\left(1\right)}DG\xi_{{⊥}_{i}}^{\left(m\right)}+\sigma_{inter}\Gamma \sum_{p=m-1>0}^{p=m+1\leq Q}\left[\xi_{{⊥}_{i}}^{\left(p\right)}-\xi_{{⊥}_{i}}^{\left(m\right)}\right],\\ \;i=2,\ldots ,N,\;\mathrm{and}\;m=1,2,\ldots ,Q; \end{array}$$If Q=2, i.e., the total number of layers is two, and the corresponding Laplacian matrices are $L^{\left(1\right)}$ and $L^{\left(2\right)}$. Furthermore, suppose that the intralayer connection in any one layer is globally coupled—for example, say layer-1 is globally coupled—then the eigenvalues of the corresponding Laplacian matrix $L^{\left(1\right)}$ are 0 with algebraic multiplicity 1 and *N* with algebraic multiplicity N−1. Then, in this scenario, the master stability Equation (Equation 11) can be fully decoupled into a low dimensional form, by projecting the transverse error components onto the basis of eigenvectors of ${\tilde{L}}^{\left(2\right)}$, which eventually gives the decoupled master stability equation as
(13)$$\begin{array}{c} {\dot{\xi }}_{{⊥}_{i}}^{\left(1\right)}=JF\left(V_{1,s},\phi_{1}\right)\xi_{{⊥}_{i}}^{\left(1\right)}-\sigma_{intra}NDG\xi_{{⊥}_{i}}^{\left(1\right)}+\sigma_{inter}\Gamma \left[\xi_{{⊥}_{i}}^{\left(2\right)}-\xi_{{⊥}_{i}}^{\left(1\right)}\right],\\ {\dot{\xi }}_{{⊥}_{i}}^{\left(2\right)}=JF\left(V_{2,s},\phi_{2}\right)\xi_{{⊥}_{i}}^{\left(2\right)}-\sigma_{intra}\lambda_{i}^{\left(2\right)}DG\xi_{{⊥}_{i}}^{\left(2\right)}+\sigma_{inter}\Gamma \left[\xi_{{⊥}_{i}}^{\left(1\right)}-\xi_{{⊥}_{i}}^{\left(2\right)}\right]; \end{array}$$Using a similar concept as in 2, we can infer that if the connectivity structures of the layers of our multiplex framework are such that the Laplacian matrix of each layer is either $L^{\left(1\right)}$ or $L^{\left(2\right)}$, and $L^{\left(1\right)}$ is the Laplacian matrix associated with globally connected networks, then the master stability equation can be fully decoupled into a lower dimensional form. For example, we say that the first $q_{1}$ number of layers have an identical Laplacian matrix $L^{1}$, and that $q_{2}\left(=Q-q_{1}\right)$ number of layers have identical Laplacian matrices $L^{\left(2\right)}$. Then, following 2, the master stability equation can be represented as
(14)$$\begin{array}{c} {\dot{\xi }}_{{⊥}_{i}}^{\left(m\right)}=JF\left(V_{m,s},\phi_{m}\right)\xi_{{⊥}_{i}}^{\left(m\right)}-\sigma_{intra}NDG\xi_{{⊥}_{i}}^{\left(m\right)}+\sigma_{inter}\Gamma \sum_{p=m-1>0}^{p=m+1}\left[\xi_{{⊥}_{i}}^{\left(p\right)}-\xi_{{⊥}_{i}}^{\left(m\right)}\right],\\ \;m=1,2,\cdots ,q1,\\ {\dot{\xi }}_{{⊥}_{i}}^{\left(m\right)}=JF\left(V_{m,s},\phi_{m}\right)\xi_{{⊥}_{i}}^{\left(m\right)}-\sigma_{intra}\lambda_{i}^{\left(2\right)}DG\xi_{{⊥}_{i}}^{\left(m\right)}+\sigma_{inter}\Gamma \sum_{p=m-1}^{p=m+1\leq Q}\left[\xi_{{⊥}_{i}}^{\left(p\right)}-\xi_{{⊥}_{i}}^{\left(m\right)}\right],\\ \;m=q1+1,2,\cdots ,Q; \end{array}$$When none of the intralayer connectivity structure is globally coupled but the Laplacian matrices $L^{\left(1\right)}$ and $L^{\left(2\right)}$ are commutative with each other, then also the master stability equation can be reduced to a lower dimensional form, because the commutative Laplacian matrices share the same set of the basis of eigenvectors that diagonalizes them. Hence, the reduced master stability equation becomes
(15)$$\begin{array}{c} {\dot{\xi }}_{{⊥}_{i}}^{\left(m\right)}=JF\left(V_{m,s},\phi_{m}\right)\xi_{{⊥}_{i}}^{\left(m\right)}-\sigma_{intra}\lambda_{i}^{\left(1\right)}DG\xi_{{⊥}_{i}}^{\left(m\right)}+\sigma_{inter}\Gamma \sum_{p=m-1>0}^{p=m+1}\left[\xi_{{⊥}_{i}}^{\left(p\right)}-\xi_{{⊥}_{i}}^{\left(m\right)}\right],\\ \;m=1,2,\cdots ,q1,\\ {\dot{\xi }}_{{⊥}_{i}}^{\left(m\right)}=JF\left(V_{m,s},\phi_{m}\right)\xi_{{⊥}_{i}}^{\left(m\right)}-\sigma_{intra}\lambda_{i}^{\left(2\right)}DG\xi_{{⊥}_{i}}^{\left(m\right)}+\sigma_{inter}\Gamma \sum_{p=m-1}^{p=m+1\leq Q}\left[\xi_{{⊥}_{i}}^{\left(p\right)}-\xi_{{⊥}_{i}}^{\left(m\right)}\right],\\ \;m=q1+1,2,\cdots ,Q. \end{array}$$

Therefore, in the above discussed scenarios, for the stability of an intralayer synchronization state, it is sufficient to check the stability of the Qd-dimensional (N−1) decoupled transverse error dynamics, instead of the Qd(N−1)-dimensional coupled system. Then, the intralayer synchronization state is locally asymptotically stable if the maximum Lyapunov exponent of these Qd-dimensional systems becomes negative.

#### 3.1.3. Simultaneous Achievement of Layer-Wise Synchronization

This subsection delves into the concurrent achievement of synchronization across all the layers. In other words, what we will try to show is that when the layers are connected to each other (i.e., interlayer coupling strength $\sigma_{inter}\neq 0$), then, for a specific value of intralayer strength $\sigma_{intra}$, either all the layers are in a state of desynchrony or synchrony. To prove this, here we will go through the method of contradiction.

Therefore, we start with the consideration that for specific values of $\sigma_{intra}$ and $\sigma_{inter}$ all the layers are in a state of synchrony, except for a particular layer. For the sake of definiteness, we say that layer-*l* is in a desynchronized state, i.e., $V_{m,1}=V_{m,2}=\cdots =V_{m,N}$, for all *m* but *l*. Then, at least two nodes, *i* and *j*, exist in layer-*l*, such that
(16)$$\begin{array}{c} V_{l,i}\neq V_{l,j}\;\mathrm{and}\;{\dot{V}}_{l,i}\neq {\dot{V}}_{l,j}. \end{array}$$
Now, in layer-(l+1), the evolution of the *i*th and *j*th nodes is given by
(17)$$\begin{array}{c} {\dot{V}}_{l+1,i}=F\left(V_{l+1,i},\phi_{l+1}\right)+\sigma_{intra}\sum_{k=1}^{N}C_{ik}^{\left(l+1\right)}G\left(V_{l+1,i},V_{l+1,k}\right)\\ \;\;\;\;\;\;\;\;\;\;\;\;\;\;\;\;\;\;\;\;+\sigma_{inter}\Gamma \sum_{p=l>0}^{p=(l+2)\leq Q}\left[V_{p,i}-V_{l+1,i}\right], \end{array}$$
and
(18)$$\begin{array}{c} {\dot{V}}_{l+1,j}=F\left(V_{l+1,j},\phi_{l+1}\right)+\sigma_{intra}\sum_{k=1}^{N}C_{jk}^{\left(l+1\right)}G\left(V_{l+1,j},V_{l+1,k}\right)\\ \;\;\;\;\;\;\;\;\;\;\;\;\;\;\;\;\;\;\;\;+\sigma_{inter}\Gamma \sum_{p=l>0}^{p=(l+2)\leq Q}\left[V_{p,j}-V_{l+1,j}\right]. \end{array}$$
According to our assumption, layer-(l+1) is in synchrony; therefore, for the *i*th and *j*th nodes, $V_{l+1,i}=V_{l+1,j}$. Now, using relation (Equation 16), from Equations (Equation 17) and (Equation 18), we find that ${\dot{V}}_{l+1,i}\neq {\dot{V}}_{l+1,j}$, due to the fact that the last term of both equations becomes unequal, which contradicts the fact that layer-(l+1) is in a state of synchrony, and the desynchrony in layer-*l* has spread in layer-(l+1).

Similarly, one can ascertain that the desynchronized motion in layer-*l* propagates into layer-(l−1). Thus, the propagation of asynchronous motion in layer-*l* can be observed in the adjacent layers-(l+1) and (l−1). Then, the asynchronous movements of adjacent layers (l−1), *l*, and (l+1) are likely to transmit to layers (l−2) and (l+2) in a brief period of time. Thus, in the multiplex framework (Equation 1), it can be observed that each layer becomes desynchronized in rapid succession within a brief timeframe, due to the existence of a single desynchronized layer. Therefore, one can conclude that asynchronous and synchronous layers cannot coexist in the multiplex network (Equation 1) when the layers are connected to each other (i.e., $\sigma_{inter}\neq 0$), and synchrony occurs in each layer concurrently.

The obtained result, concerning the simultaneous emergence of synchronization in all the layers, plays a crucial role in our study, as changes made in one layer will affect the occurrence of synchronization in all the other layers. Throughout the next section, we will investigate how mismatch, in terms of system parameters $\phi_{m}$ in one layer, influences the emergence of intralayer synchrony in the whole multiplex network (Equation 1).

### 3.2. Numerical Results

Intralayer synchronization in the multiplex framework (Equation 1) implies the occurrence of complete synchrony in each layer. Therefore, in order to quantify the intralayer synchronous state, we introduce the intralayer synchronization error as $E_{intra}=\frac{1}{Q}\sum_{m=1}^{Q}E_{m}$, where
(19)$$\begin{array}{c} E_{m}={\langle {\left(\sum_{i,j=1}^{N}\frac{∥V_{m,j}\left(t\right)-V_{m,i}\left(t\right)∥}{N(N-1)}\right)}^{\frac{1}{2}}\rangle }_{T} \end{array}$$
quantifies the average complete synchronization error inside layer-*m*. Here, ‖ ‖ represents the standard Euclidean norm, and *T* is an adequately large interval of time over which the synchronization error $E_{m}$ is averaged. $E_{m}=0$ indicates the emergence of complete synchrony in layer-*m*, whereas a nonzero value corresponds to asynchrony. Consequently, the zero (nonzero) value of $E_{intra}$ signifies the achievement of intralayer synchronization (desynchronization) in the multiplex network.

For the sake of simplicity, we first consider Q=2, i.e., the total number of layers in the multiplex network is assumed to be two, and the connectivity structures of both the layers are represented by an Erdős–Rényi (ER) [53] random network, with N=100 and edge-generating probability $p_{rand}$. The isolated node dynamics in both layers are represented by chaotic Rössler oscillators [54], with the nodes in a particular layer to be identical, while they are different in different layers, in terms of system parameters. We consider the synchronization noninvasive intralayer coupling in all the layers to be linear diffusive through the first state variable of the Rössler oscillator, and the interlayer coupling is assumed to be through the second state variable. The evolution dynamics of the multiplex network can then be expressed as
(20)$$\begin{array}{c} {\dot{x}}_{1,j}=-\omega_{1}y_{1,j}-z_{1,j}+\sigma_{intra}\sum_{k=1}^{N}C_{jk}^{\left(1\right)}\left(x_{1,k}-x_{1,j}\right),\\ {\dot{y}}_{1,j}=\omega_{1}x_{1,j}+ay_{1,j}+\sigma_{inter}\left(y_{2,j}-y_{1,j}\right),\\ {\dot{z}}_{1,j}=b+\left(x_{1,j}-c\right)z_{1,j},\\ {\dot{x}}_{2,j}=-\omega_{2}y_{2,j}-z_{2,j}+\sigma_{intra}\sum_{k=1}^{N}C_{jk}^{\left(2\right)}\left(x_{2,k}-x_{2,j}\right),\\ {\dot{y}}_{2,j}=\omega_{2}x_{2,j}+ay_{2,j}+\sigma_{inter}\left(y_{1,j}-y_{2,j}\right),\\ {\dot{z}}_{2,j}=b+\left(x_{2,j}-c\right)z_{2,j},j=1,2,\ldots ,N, \end{array}$$
where the parameter values are fixed at a=0.2, b=0.2, and c=5.7, for which each individual oscillator exhibits chaotic dynamics. The isolated node dynamics of the two layers are different from one other, in terms of the parameters $\omega_{1}$ and $\omega_{2}$, where $\omega_{1}=\omega$ and $\omega_{2}=\omega +\Delta \omega$, with Δω being the parameter of mismatch between the layers. Here, we fix the value of ω=1, and the range of Δω has been chosen in such a way that each individual node in the second layer shows chaotic time evolution. Notably, the *x* coupling induces synchronization of the bounded type in a single-layer network of Rössler oscillators [50]. Therefore, we choose this coupling configuration to investigate how parameter mismatch among the layers affects the region of synchronization within each layer, i.e., whether the bounded region of synchronization expands or shrinks due to the presence of mismatch among the layers.

As the main focus of the present study is to investigate the effect of parameter mismatch on the occurrence of intralayer synchronization in a multiplex framework, we here take the connectivity structure of both layers to be identical, i.e., $C^{\left(1\right)}=C^{\left(2\right)}$, so that parameter mismatch will only play the role of a difference maker in our investigation. Therefore, we take the connectivity structure of both layers to be a random network with N=100 nodes and edge-generating probability $p_{rand}=0.1$, and we integrate the system (Equation 20) for a period of $3\times 10^{5}$ time steps, with the integration step size δt=0.01, and the last $10^{5}$ time units being taken for calculating the average synchronization error.

In order to study the effect of parameter mismatch on the emergence of intralayer synchronization in a multiplex network, we first investigate the layer-wise synchronization, by evaluating the synchronization errors $E_{1}$ and $E_{2}$ associated with layer-1 and layer-2, for a fixed value of mismatch parameter Δω=0.1 and for three different values of interlayer coupling strength $\sigma_{inter}$. The corresponding results are depicted in Figure 2. The solid and dashed curves represent the variation of synchronization error $E_{1}$ and $E_{2}$, respectively, as a function of intralayer coupling strength $\sigma_{intra}$. When the layers are not connected, i.e., when $\sigma_{inter}=0$ (red curves), synchronization in layer-1 and layer-2 emerges at two different critical intralayer coupling strengths: $\sigma_{intra}^{*}\approx 0.06$ and $\sigma_{intra}^{*}\approx 0.055$, respectively. Due to the absence of interlayer connection and to two different parameter values, the two layers achieve synchrony at two different values of $\sigma_{intra}$ and, furthermore, layer-2 achieves synchrony at a comparably lower critical coupling, as compared to layer-1. In the presence of interlayer connections, i.e., $\sigma_{inter}\neq 0$, both layers achieve synchrony concurrently. The blue and green curves in Figure 2 illustrate the variation of the synchronization errors $E_{1}$ and $E_{2}$ for sufficiently lower and sufficiently larger interlayer coupling strengths $\sigma_{inter}=0.001$ and $\sigma_{inter}=0.1$, respectively. For a smaller value of $\sigma_{inter}$, both layers achieve synchrony at exactly the same critical coupling, $\sigma_{intra}^{*}\approx 0.058$, while, for $\sigma_{inter}=0.1$, synchrony in both layers emerges at an adequately smaller value of intralayer coupling strength, $\sigma_{intra}^{*}\approx 0.01$. This affirms our finding that all the layers of multiplex networks synchronize simultaneously when coupled through interlayer connections. Furthermore, from Figure 2, we can conclude that the presence of parameter mismatch and sufficient interlayer coupling between the layers induces an enhancement in intralayer synchronization in multiplex networks.

Proceeding, we evaluate the intralayer synchronization error $E_{intra}$ as a function of $\sigma_{intra}$ for different values of mismatch parameter Δω and a fixed value of the interlayer coupling strength $\sigma_{inter}=0.1$. Figure 3a delineates the corresponding results, where red, blue, and magenta curves illustrate the intralayer synchronization transition for three different values of mismatch parameter Δω=0, Δω=0.02, and Δω=0.1, respectively. Here, we observe that the intralayer synchronization occurs in a bounded interval I=[α,β], where α(β) indicates the critical intralayer coupling for the transition from desynchrony to synchrony (synchrony to desynchrony). For Δω=0, i.e., when both the layers are identical, the intralayer synchronization error becomes 0 within the range of interval I≈[0.055,0.215]. In the presence of parameter mismatch (Δω=0.02), the intralayer synchronization occurs for a comparably larger interval of intralayer coupling strength $\sigma_{intra}\in I\approx \left[0.041,0.225\right]$. For larger values of parameter mismatch (Δω=0.1), intralayer synchrony emerges even in a wider range of coupling strength I≈[0.01,0.24]. Therefore, intralayer synchrony on multiplex networks can emerge more easily, due to the introduction of parameter mismatch among the layers, than on a multiplex network with identical layers. In other words, due to the parameter mismatch in one layer, the other layer can achieve synchrony at those values of intralayer coupling strengths at which synchronization is not possible with identical parameter values. Hence, we can conclude that parameter mismatch in one layer induces synchronization in another one.

To validate the acquired result, we proceed through the linear stability analysis of the intralayer synchronous solution, by evaluating the maximum Lyapunov exponent transverse to the synchronization manifold, as discussed in Section 3.1.2. Therefore, we evaluate the maximum transverse Lyapunov exponent $\Lambda_{intra}$, whose negative values for varying coupling strengths indicate the region of a stable synchronous solution. As we have assumed the connectivity structure in both layers to be identical, the calculation of $\Lambda_{intra}$ can be done by solving the master stability Equation (Equation 12). In Figure 3b, we plot the curves of $\Lambda_{intra}$ as a function of the intralayer coupling strength $\sigma_{intra}$ for the same set of values taken for the evaluation of $E_{intra}$ in the upper panel. The red, blue, and magenta curves illustrate the variation of $\Lambda_{intra}$ for three different values of mismatch parameter Δω=0,0.02,and0.1, respectively. As observed, the curves of $\Lambda_{intra}$ remain negative in the same bounded region I=[α,β], for which the value of the synchronization error is 0. Thus, our observation regarding the enhancement in the region of occurrence of intralayer synchronization with the introduction of a parameter mismatch between the layers is validated analytically, using the linear stability analysis.

Thereafter, to scrutinize the complete scenario of intralayer synchronization in a wider range of parameter values, we evaluate $E_{intra}$ by simultaneously varying the intralayer and interlayer couplings, $\sigma_{intra}$ and $\sigma_{inter}$, respectively, for two different values of mismatch parameter Δω. The corresponding result is illustrated in Figure 4, where the color bars indicate the variation of $E_{intra}$, with the deep blue region representing the domain of synchronization. In the absence of mismatch (Δω=0), i.e., when both the layers are identical, intralayer synchronization emerges within a bounded region of coupling strength, where the left and right bound of the interval indicates the critical intralayer coupling strengths for the transition from desynchrony to synchrony and from synchrony to desynchrony, respectively. Furthermore, these left and right critical points for the emergence of synchrony are almost identical, independent of the value of $\sigma_{inter}$, i.e., the occurrence of intralayer synchrony for identical layers is almost independent of the interlayer connections (see Figure 4a). On the other hand, in Figure 4b, when a mismatch is introduced (i.e., Δω=0.1), the area of synchrony increases along with the increasing value of interlayer coupling $\sigma_{inter}$. Here, the critical coupling for the transition from desynchrony to synchrony decreases with increasing values of $\sigma_{inter}$, up to $\sigma_{inter}\approx 0.055$. Beyond that, the critical coupling for the achievement of intralayer synchronization is almost identical, independent of the value of $\sigma_{inter}$. In addition, the critical coupling for the transition from synchrony to desynchrony increases with increasing $\sigma_{inter}$. Therefore, due to the introduction of a mismatch between the layers, the multiplex network can achieve intralayer synchrony for those pairs of $\left(\sigma_{intra},\sigma_{inter}\right)$, where the intralayer synchronization is forbidden for multiplex network with identical layers, or, in other words, mismatch introduced in one layer induces synchronization in another, because of the simultaneous occurrence of layer-wise synchrony. This eventually results in an enhancement of the area of intralayer synchronization.

Until now, the results we have discussed have been for multiplex networks with fixed intralayer connectivity structure, i.e., the value of edge-joining probability $p_{rand}$ is fixed. But, $p_{rand}$ is one of the most important parameters in our study, as it indicates how densely the nodes within the layers are connected to one other. Therefore, we now investigate the combined effect of mismatch and layer-wise connectivity density. In Figure 5, we evaluate the variation of the intralayer synchronization error, by simultaneously varying $\sigma_{intra}$ and $p_{rand}$ for the identical parameter, i.e., Δω=0 (Figure 5a) and mismatch parameter with Δω=0.1 (Figure 5b), while keeping the interlayer coupling strength fixed at $\sigma_{inter}=0.1$. As observed, the region of synchronization enhances in the presence of the mismatch parameter, compared to the scenario where no mismatch among the layers is considered: in other words, due to the introduction of mismatch among the layers, intralayer synchronization emerges for those connectivity probabilities at which the multiplex network with identical layers is unable to achieve synchrony.

Now, in order to show the generality of the acquired results in the framework of multiplex networks, we consider Q=3, i.e., the multiplex network is composed of three layers organized in a chain, and the connectivity structure of all the three layers are represented, as previously, by an Erdos–Rényi (ER) random network, with N=100 and edge-generating probability $p_{rand}$. The isolated node dynamics in both layers are represented by chaotic Rössler oscillators, with the nodes in a particular layer being identical, while they are different in different layers, in terms of system parameters. Once again, we consider the synchronization noninvasive intralayer coupling in all the layers to be linear diffusive through the first state variable of the Rössler oscillator, and the interlayer coupling is assumed to be through the second state variable. Then, the evolution dynamics of the multiplex network can be represented as
(21)$$\begin{array}{c} {\dot{x}}_{1,j}=-\omega_{1}y_{1,j}-z_{1,j}+\sigma_{intra}\sum_{k=1}^{N}C_{jk}^{\left(1\right)}\left(x_{1,k}-x_{1,j}\right),\\ {\dot{y}}_{1,j}=\omega_{1}x_{1,j}+ay_{1,j}+\sigma_{inter}\left(y_{2,j}-y_{1,j}\right),\\ {\dot{z}}_{1,j}=b+\left(x_{1,j}-c\right)z_{1,j},\\ {\dot{x}}_{2,j}=-\omega_{2}y_{2,j}-z_{2,j}+\sigma_{intra}\sum_{k=1}^{N}C_{jk}^{\left(2\right)}\left(x_{2,k}-x_{2,j}\right),\\ {\dot{y}}_{2,j}=\omega_{2}x_{2,j}+ay_{2,j}+\sigma_{inter}\left(y_{1,j}+y_{3,j}-2y_{2,j}\right),\\ {\dot{z}}_{2,j}=b+\left(x_{2,j}-c\right)z_{2,j},\\ {\dot{x}}_{3,j}=-\omega_{3}y_{3,j}-z_{3,j}+\sigma_{intra}\sum_{k=1}^{N}C_{jk}^{\left(3\right)}\left(x_{3,k}-x_{3,j}\right),\\ {\dot{y}}_{2,j}=\omega_{3}x_{3,j}+ay_{3,j}+\sigma_{inter}\left(y_{2,j}-y_{3,j}\right),\\ {\dot{z}}_{3,j}=b+\left(x_{3,j}-c\right)z_{3,j},\;\;\;j=1,2,3\ldots ,N, \end{array}$$
where the parameter values are fixed at a=0.2, b=0.2, and c=5.7, for which each individual oscillator exhibits chaotic dynamics. From Equation (Equation 21), it is clear that layer-2 is adjacent to both layer-1 and layer-3, while layer-1 and layer-3 are connected through the interlayer links only by layer-2. The isolated node dynamics of the layers are different from one other, in terms of the intrinsic parameters $\omega_{1}$, $\omega_{2}$, and $\omega_{3}$, where $\omega_{1}=\omega$, $\omega_{2}=\omega +\Delta \omega$, and $\omega_{3}=\omega +2\Delta \omega$, with Δω being the parameter of mismatch between the layers. Here, we fix the value of ω=1, and the range of Δω has been chosen in such a way that each individual node in the second and third layers exhibits chaotic time evolution.

To study the effect of parameter mismatch on the occurrence of intralayer synchronization in the three-layered multiplex network (Equation 21), we again take the connectivity structure of all the layers to be identical, i.e., $C^{\left(1\right)}=C^{\left(2\right)}=C^{\left(3\right)}$, so that the mismatch parameter will only play the role of difference maker in our investigation. Therefore, we consider the connectivity structure of each layer to be a random network, with N=100 nodes and edge-generating probability $p_{rand}=0.1$, and we evaluate the intralayer synchronization error $E_{intra}$ as a function of $\sigma_{intra}$ for different values of Δω, and a fixed value of interlayer coupling strength $\sigma_{inter}=0.1$. The corresponding results are delineated in Figure 6a, where the red, blue, and magenta curves represent the intralayer synchronization transition for three different values of mismatch parameter Δω=0, Δω=0.05, and Δω=0.1, respectively. It can be seen that the intralayer synchronization occurs in a bounded interval, I=[α,β], for all the three values of the mismatch parameter. However, the region of synchronization widens as the amount of mismatch increases, through increasing the value of Δω. We further verify the acquired result by evaluating the maximum Lyapunov exponent $\Lambda_{intra}$ transverse to the intralayer synchronization manifold $S^{intra}$. In Figure 6b, we plot the curves of $\Lambda_{intra}$ as a function of $\sigma_{intra}$ for the same set of parameter values for which $E_{inta}$ has been calculated. Clearly, the curves of $\Lambda_{intra}$ remain negative in the same bounded region I for which the value of $E_{intra}$ is 0. Thus, our observation regarding the enhancement in the region of occurrence of intralayer synchronization, with the introduction of a parameter mismatch between the layers of the multiplex network, holds true also for the three-layered multiplex network.

## 4. Conclusions

We have investigated the intralayer synchronization phenomenon in multiplex networks where the layers are different from one other, in terms of parameter mismatch in the intrinsic dynamics of unitary components, while the intrinsic dynamics of each unitary component of a particular layer are identical. Based on the assumption of synchronization noninvasive coupling functions (which is, however, necessary for the emergence of an intralayer synchronous solution), we derived a necessary condition for the referred solution to be stable, using the master stability function approach. We further uncovered that, despite the layers being nonidentical, all the layers of the multiplex network synchronize simultaneously when connected through interlayer connections. Using the paradigmatic chaotic Rössler oscillators to describe the intrinsic dynamics of individual components of the layers, we showed that the layers can achieve synchrony at comparably lower values of coupling strengths, with an increasing value of mismatch parameter between the layers. In addition, we found that the area of intralayer synchronization becomes wider with the increasing value of the mismatch parameter, which was analytically validated by performing the stability analysis of the synchronous solution. Our study also revealed that, due to the introduction of a mismatch between the layers, through changing the parameter values in one layer, the other layer can achieve synchrony at those values of coupling strengths at which synchronization is forbidden when the layers are identical, i.e., heterogeneity enhances synchrony.

Thus, we have provided a significant contribution to the investigation of intralayer synchronization in multiplex networks subject to layer mismatch, even though many areas of further research are still unexplored. In particular, here, we have considered the heterogeneity among different layers, by means of constant parameter mismatch in the intrinsic dynamics of individual elements of the layers. Therefore, consideration of adaptive mismatch procedure with various intralayer and interlayer coupling schemes is of special interest for future research. We here provided the numerical results only for a small mismatch in system parameters, which show that parameter mismatch enhances intralayer synchronization. However, the question remains: does it hold true even with a very large mismatch in parameters? Therefore, another intriguing area of future research is to investigate the effect of a sufficiently large parameter mismatch on the emergence of intralayer synchronization. We believe that our study can provide a better understanding of the impact of parameter mismatch among the layers of multiplex networks in the emergence of different kinds of synchronization behavior.

## Figures and Tables

**Figure 1 entropy-25-01083-f001:**
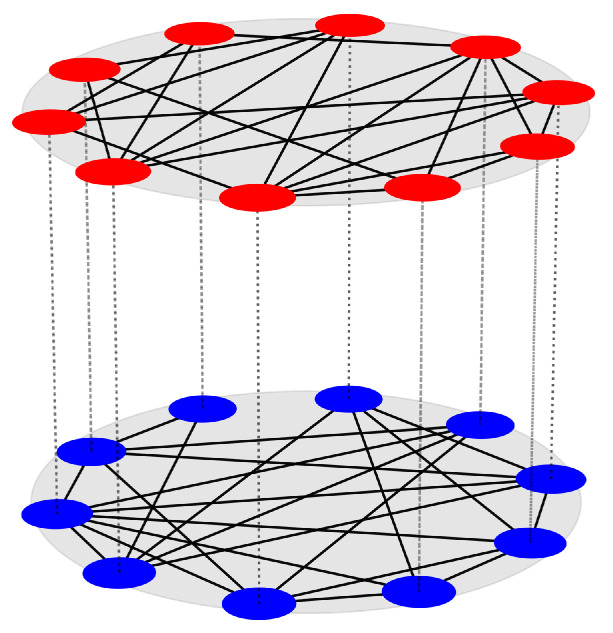
Schematic representation of a Q=2 layered multiplex network with N=10 nodes in each layer. The nodes in each layer are identical to one other, while being different for different layers, by means of parameter mismatch in the local dynamics. To illustrate this, the nodes in layer-1 are colored in red, and the nodes in layer-2 are colored in blue. Solid black lines portray the intralayer connections, while dashed black lines show the replica-wise interlayer connections.

**Figure 2 entropy-25-01083-f002:**
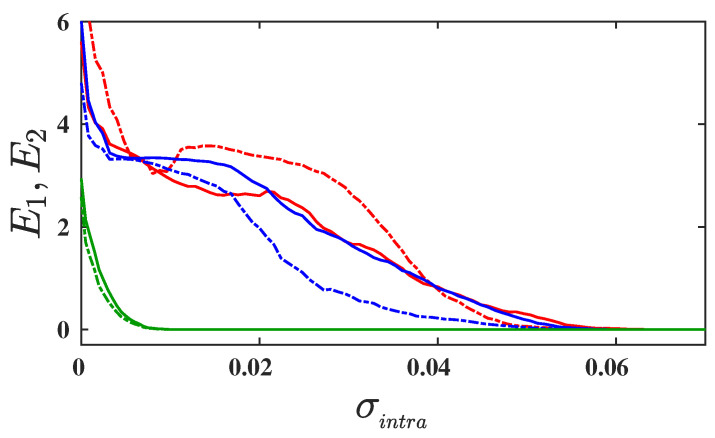
Synchronization errors $E_{1}$ and $E_{2}$, corresponding to layer-1 and layer-2, are depicted by solid and dashed curves as a function of intralayer coupling strength $\sigma_{intra}$ for three different values of interlayer coupling strengths: $\sigma_{inter}=0$ (in red); $\sigma_{inter}=0.01$ (in blue); and $\sigma_{inter}=0.1$ (in green). The mismatch parameter is fixed at Δω=0.1. The connectivity probability of both layers is fixed at $p_{rand}=0.1$. All the curves are obtained by taking 10 different network realizations and initial conditions, which are drawn from the phase space of isolated node dynamics.

**Figure 3 entropy-25-01083-f003:**
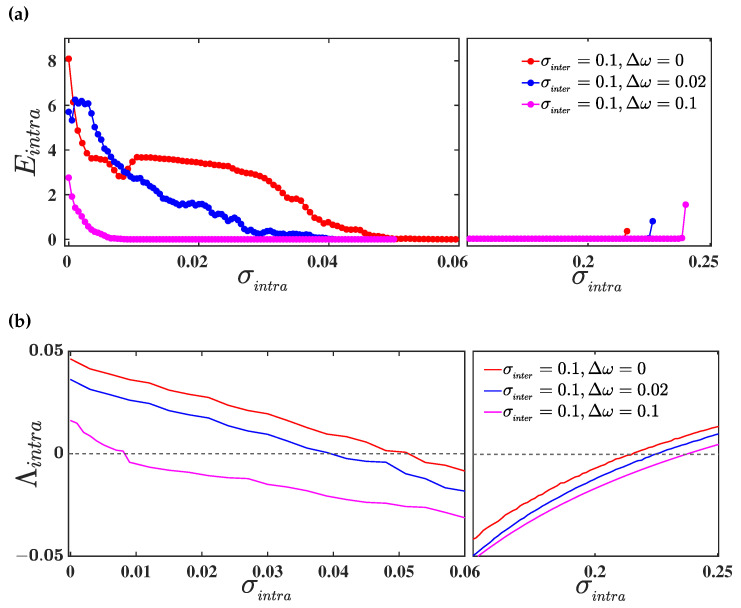
(**a**) Variation of intralayer synchronization error as a function of $\sigma_{intra}$ for three different values of mismatch parameter Δω=0 (in red), Δω=0.02 (in blue), and Δω=0.1 (in magenta), respectively. The interlayer coupling strength is fixed at $\sigma_{inter}=0.1$; (**b**) the maximum Lyapunov exponent $\Lambda_{intra}$ as a function of $\sigma_{intra}$ for the same set of mismatch parameter values taken in the upper panel. The dashed horizontal black line represents the 0 line. In both the upper and lower panel, the network connectivity probability is taken to be $p_{rand}=0.1$.

**Figure 4 entropy-25-01083-f004:**
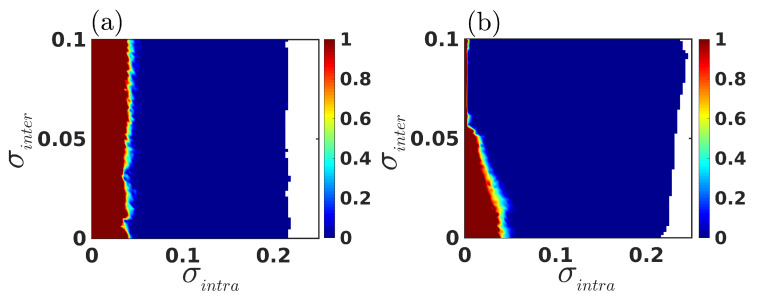
Variation of intralayer synchronization error $E_{intra}$ as a function of $\sigma_{intra}$ and $\sigma_{inter}$ for two different values of mismatch parameter Δω=0 (**a**) and Δω=0.1 (**b**), respectively. For both the subfigures, in the white region the system becomes unbounded. The color bars indicate the variation of intralayer synchronization error $E_{intra}$, where the 0 value of $E_{intra}$ represents the emergence of intralayer synchrony. The connectivity probability in both the layers is fixed at $p_{rand}=0.1$.

**Figure 5 entropy-25-01083-f005:**
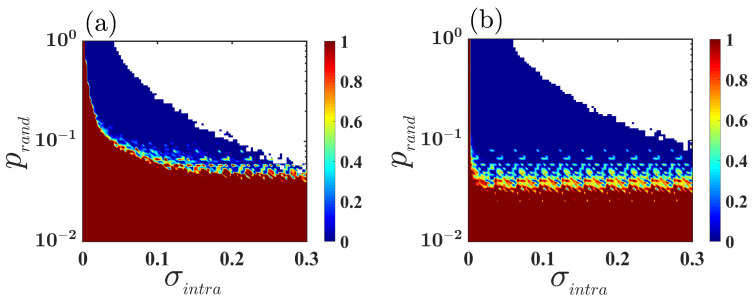
Variation of intralayer synchronization error $E_{intra}$ as a function of intralayer coupling strength $\sigma_{intra}$ and edge-generating probability $p_{rand}$ for two different values of mismatch parameter Δω=0 (**a**) and Δω=0.1 (**b**), respectively. In both the subfigures, the region in white indicates the unbounded region. The color bars indicate the variation of intralayer synchronization error $E_{intra}$, where the 0 value of $E_{intra}$ represents the emergence of intralayer synchrony. The interlayer coupling strength is fixed at $\sigma_{inter}=0.1$.

**Figure 6 entropy-25-01083-f006:**
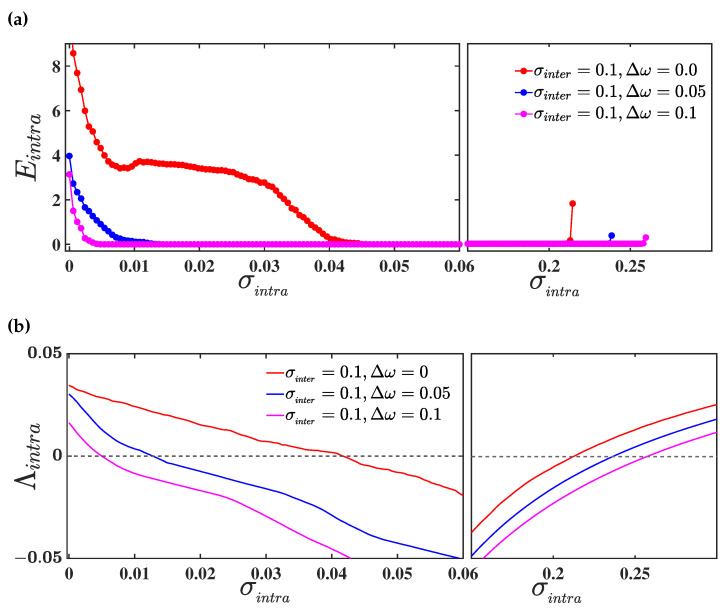
(**a**) Variation of intralayer synchronization error as a function of $\sigma_{intra}$ for three different values of mismatch parameter Δω=0 (in red), Δω=0.05 (in blue), and Δω=0.1 (in magenta), respectively. The interlayer coupling strength is fixed at $\sigma_{inter}=0.1$; (**b**) the maximum Lyapunov exponent $\Lambda_{intra}$ as a function of $\sigma_{intra}$ for the same set of mismatch parameter values taken in the upper panel. The dashed horizontal black line represents the 0 line. In both the upper and lower panel, the network connectivity probability is taken to be $p_{rand}=0.1$.

## Data Availability

The paper itself contains all the information required to assess the conclusions. Additional data related to this paper may be requested from the corresponding author.

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
