# Peer review of "Synchronization Induced by Layer Mismatch in Multiplex Networks"

_entropy, 2023, doi:10.3390/e25071083_

Round 1

Reviewer 1 Report

In this manuscript, Anwar et al. introduce an analytical framework to assess the synchronization in the multiplex network induced by mismatched layers. The paper represents a solid analytical work supported by numerical simulations. Overall, I would like to emphasize the rigor and scientific soundness of the manuscript.

However, I have two minor comments that should be addressed before acceptance.

1.    In the governing equation (1), the authors introduce intralayer coupling through the function G, however, the interlayer coupling is given explicitly by the difference of the states of respective nodes in different layers. Is there any reason behind that?
2.    In Eq. (20), the notation of parameters a_1, a_2, and a_3 is a bit confusing, since the subscripts in this equation indicate the layer's number. Consider using the classical notation of these parameters like a_1=a, a_2=b, and a_3=c.

Reviewer 2 Report

This manuscript is well addressed, and multi-layer network is suitable to describe the collective behaviors of complex systems with distinct diversity.  Similar objects or agaents can be clustered into the same layer or local area in a layer network, and interplay between different layers can be handled under coupling assumptions.  Some questions can be clarfied for a miner revision before final positive recommendation.

1.  What is heterogeneity in your network? How to describe or characterize a heterogeneity? What is the formation mechanim for creation of heterogeneity in the network.  As mentioned in the recent work[https://link.springer.com/article/10.1007/s11431-022-2188-2; https://doi.org/10.1016/j.cnsns.2023.107127], gradient distrubition and collection of energy in local area will develop heterogenetiy in the network. Please clarify your comments.

2. For multi-layer network, inner coupling in the same layer and interaction between different layes have different impacts on the synchronization stability,  in presence of heterogeneity, which kinds of coupling plays main decision on the synchronization approach?

3, Is it possible to control the coupling intensity in adaptive way?

Reviewer 3 Report

see the attachment

The logic description and language quality shoul dbe improved.

Round 2

Reviewer 1 Report

The authors addressed all comments. I support publishing paper in its present form.

Reviewer 2 Report

My questions have been considered well, this version can be suitable for possible acceptance.

Reviewer 3 Report

The paper has been satisfactorily revised. I suggest to accept it.